# Constructing, validating, and updating machine learning models to predict survival in children with Ebola Virus Disease

**Alicia E. Genisca**[1], **Kelsey Butler**[2], **Monique Gainey**[3], **Tzu-Chun Chu**[4], **Lawrence Huang**[5], **Eta N. Mbong**[6], **Stephen B. Kennedy**[7], **Razia Laghari**[6], **Fiston Nganga**[6], **Rigobert F. Muhayangabo**[6], **Himanshu Vaishnav**[8], **Shiromi M. Perera**[9], **Moyinoluwa Adeniji**[5], **Adam C. Levine**[1‡], **Ian C. Michelow**[10‡], **Andrés Colubri**[2‡]*

**1** The Warren Alpert School of Medicine, Brown University, Providence, Rhode Island, United States of America, **2** University of Massachusetts Chan Medical School, Worcester, Massachusetts, United States of America, **3** Rhode Island Hospital, Providence, Rhode Island, United States of America, **4** University of Georgia, Athens, Georgia, United States of America, **5** Brown University, Providence, Rhode Island, United States of America, **6** International Medical Corps, Goma, Democratic Republic of Congo, **7** Ministry of Health, Monrovia, Liberia, **8** Brown Emergency Medicine, Providence, Rhode Island, United States of America, **9** International Medical Corps, Washington, Washington, United States of America, **10** Connecticut Children's Medical Center, University of Connecticut School of Medicine, Hartford, Connecticut, United States of America

☯ These authors contributed equally to this work.
‡ ACL, ICM and AC also contributed equally to this work.
* Andres.Colubri@umassmed.edu

**Data Availability Statement:** The Ebola Data Access Committee (DAC) manages and oversees all data access applications for re-use of the West Africa EDP derivation dataset, in accordance with

## Abstract

### Background

Ebola Virus Disease (EVD) causes high case fatality rates (CFRs) in young children, yet there are limited data focusing on predicting mortality in pediatric patients. Here we present machine learning-derived prognostic models to predict clinical outcomes in children infected with Ebola virus.

### Methods

Using retrospective data from the Ebola Data Platform, we investigated children with EVD from the West African EVD outbreak in 2014–2016. Elastic net regularization was used to create a prognostic model for EVD mortality. In addition to external validation with data from the 2018–2020 EVD epidemic in the Democratic Republic of the Congo (DRC), we updated the model using selected serum biomarkers.

### Findings

Pediatric EVD mortality was significantly associated with younger age, lower PCR cycle threshold (Ct) values, unexplained bleeding, respiratory distress, bone/muscle pain, anorexia, dysphagia, and diarrhea. These variables were combined to develop the newly described EVD Prognosis in Children (EPiC) predictive model. The area under the receiver operating characteristic curve (AUC) for EPiC was 0.77 (95% CI: 0.74–0.81) in the West

their Data Access Guidelines and Data Transfer Agreement. Access can be requested to the DAC from: https://www.iddo.org/ebola/data-sharing/accessing-data. The DRC validation dataset is provided as a supplementary table (S1 Data). All source code used for model construction, validation, and updating is deposited in the following repository: https://github.com/colabobio/ebola-pediatric-prognostic-model/.

**Funding:** AEG received funding for data collection from the Rhode Island Foundation [grant number 5222_20200596] and the National Institute of Allergy and Infectious Diseases [grant number R25AI140490]. The funders had no role in study design, data collection and analysis, decision to publish, or preparation of the manuscript.

**Competing interests:** The authors have declared that no competing interests exist.

Africa derivation dataset and 0.76 (95% CI: 0.64–0.88) in the DRC validation dataset. Updating the model with peak aspartate aminotransferase (AST) or creatinine kinase (CK) measured within the first 48 hours after admission increased the AUC to 0.90 (0.77–1.00) and 0.87 (0.74–1.00), respectively.

## Conclusion

The novel EPiC prognostic model that incorporates clinical information and commonly used biochemical tests, such as AST and CK, can be used to predict mortality in children with EVD.

### Author summary

Although case fatality rates remain high, there are limited data on predicting mortality in children with Ebola Virus Disease (EVD). Furthermore, challenges in predicting EVD outcomes using clinical and laboratory data highlight the need for the development and validation of pediatric predictive models. The novel EVD Prognosis in Children (EPiC) model uses clinical and biochemical information, such as AST and CK, to predict mortality in infected children. While few prognostic models or scoring systems have been developed to predict clinical outcomes of EVD, the majority of them were limited in geographical and temporal scope having been derived using data from one location. As such, the EPiC model is the first externally validated model for the prognosis of pediatric EVD using diverse datasets from geographically and temporally separate outbreaks. This model can be easily applied by bedside clinicians to assess pediatric patients at risk for death and help to allocate resources accordingly.

## Introduction

With more than 28,000 cases and 11,000 deaths throughout Guinea, Liberia, and Sierra Leone, the 2014–2016 Ebola Virus Disease (EVD) outbreak in West Africa was the largest in history [1]. Within the first nine months of this epidemic, an estimated 13.8% of all EVD infections occurred in children under the age of 15 with an estimated case fatality rate (CFR) of 73.4% [2]. More recent observational data show that CFRs from the West Africa outbreak were highest among children under 5 years of age with the highest CFR of 89% reported in Guinea [1,3–6]. Similar trends were witnessed in the second largest EVD outbreak in the Democratic Republic of the Congo (DRC) from 2018–2020. Officially declared an outbreak on August 1, 2018, the tenth EVD outbreak in the DRC had a reported 3,470 EVD cases with children accounting for more than one third of cases and one in ten cases were children under five years of age [7,8]. The index case of the 13th EVD outbreak in the DRC starting in October of 2021 was a child under 5 and more than half of the confirmed cases to date are children [9]. Such findings suggest that young children are especially vulnerable and remain at higher risk of poor outcomes than older children and adults [4–6,10–12].

Several investigators have attempted to identify clinical features associated with mortality among children with EVD [6]. While common themes emerge, there is no consensus on whether certain clinical information may accurately predict outcomes of EVD because signs and symptoms tend to be non-specific [13–15]. Clinical manifestations of EVD, such as

vomiting, diarrhea, and fatigue, are similar among both children and adults; however, there are differences in their frequency and severity [15–18]. One retrospective cohort study of children under 5 years of age reported that 25% of the EVD-confirmed patients were afebrile, while in another study children were less likely than adults to report abdominal, chest, muscle, or joint pain [1,6]. The latter finding may reflect the difficulty young children have in reporting subjective symptoms. Furthermore, common laboratory tests are frequently abnormal in children with EVD [19–21]. Such challenges in predicting EVD outcomes using clinical and laboratory data highlight the need for the development and validation of pediatric predictive models.

Ebola virus tends to have a shorter incubation period and cause more rapid disease progression in children. Therefore, developing a pediatric EVD prognostic model is critical to allow clinicians to promptly identify which children may need more intensive monitoring and interventions [6,15]. Such a model could potentially inform clinical practice by allowing clinicians to optimally allocate scarce resources. Although there are a few prognostic models or scoring systems have been developed to predict clinical outcomes of EVD, they are limited in geographical and temporal scope having been derived using data from one location [22–25]. More importantly, none of them has been rigorously externally validated, limiting their generalizability and utility, and are not pediatric specific [22–25]. Colubri et al developed and validated prognostic models using data from patients of all ages at multiple treatment sites in Sierra Leone and Liberia [25]. However, data were aggregated in 10-year age bands and similar to other studies, the model was not independently validated in a distinct region of Africa during a different outbreak [22–25]. In order to fill this gap, the aim of this study was to develop and externally validate the first pediatric-specific EVD prognostic model using diverse datasets from geographically and temporally separate outbreaks.

## Methods

### Ethics statement

The Rhode Island Hospital Institutional Review Board provided an exemption from ethical review and informed consent for this secondary analysis of de-identified data as it was not considered human subjects research.

### Study design and setting

This study used retrospective data from children presenting to Ebola treatment units (ETUs) in West Africa and the DRC. The West Africa derivation dataset was built from the Infectious Diseases Data Observatory's (IDDO) Ebola Data Platform (EDP). IDDO's EDP is the first global data repository for clinical, epidemiological, and laboratory data from patients with EVD during the 2014–2015 West Africa outbreak (specifically Liberia, Guinea and Sierra Leone) provided by the following organizations: Alliance for International Medical Action (ALIMA), International Medical Corps (IMC), Institute of Tropical Medicine Antwerp (ITM), Médecins Sans Frontières (MSF), University of Oxford, Save the Children International (SCI), who had no role in the conduct of this study [26–38].

The validation DRC dataset was derived from patients who presented at IMC's Mangina ETU during the 2018–2020 EVD outbreak in the DRC. The DRC's eastern provinces of North Kivu and Ituri served as the main catchment area for the Mangina ETU, located in North Kivu.

### Participant selection

All patients less than 18 years of age who presented to West African ETUs from June 2014 to October 2015 and to the Mangina ETU from December 2018 to January 2020 with laboratory

confirmed EVD were eligible for inclusion in the derivation and validation datasets, respectively. Patients were excluded if they had missing outcome data or if they died on the day of admission to the ETU.

## EVD triage and diagnosis

### West Africa

Since the data from Liberia, Guinea, and Sierra Leone were provided by several humanitarian aid organizations, triage procedures varied slightly from site to site. All organizations adhered to World Health Organization (WHO) diagnostic criteria and relevant national guidelines [39–42].

### DRC

All patients presenting at the IMC's Mangina ETU were screened by trained clinical staff to ensure they met the clinical case definition for suspected EVD based on WHO and MSF guidelines and in consultation with local health authorities [39–42]. If patients presented with a documented diagnosis of EVD, they were directly admitted to the ward for patients with confirmed disease. Otherwise, patients who met the case definition but had no prior testing were admitted to the ETU's ward for suspected cases, where they underwent EVD testing. If the patient's initial test was negative, they remained in the ETU until 72 hours had passed, and a second EVD test was negative, in which case they were discharged. Patients with a positive test result were moved to the "confirmed" ward for further management [43,44].

### Laboratory methods

All PCR cycle threshold (Ct) values presented in this study are based on RT-PCR of the same Zaire *ebolavirus* nucleoprotein locus using standardized RNA extraction procedures [43,44]. A Ct greater than 40 was considered negative in all cases.

### West Africa

Data were provided by several humanitarian aid organizations and consequently laboratory methods differed slightly among treatment sites.

### DRC

DRC's ETUs received all patients from the surrounding catchment areas some of whom may or may not have had laboratory confirmed EVD in the community or other test facility prior to arrival. Patients were diagnosed with EVD with a RT-PCR (GeneXpert) blood assay using plasma. Blood chemistry tests were completed at point of care using Piccolo Amlyte 13, which determined levels of glucose, creatinine (CRE), albumin (ALB), aspartate aminotransferase (AST), alanine aminotransferase (ALT), amylase (AMY), potassium, C-reactive protein (CRP), total urea nitrogen (BUN), total bilirubin (TBL), creatine kinase (CK), sodium, and calcium.

### Descriptive data analysis

If the Shapiro-Wilk test for normality indicated that data were not normally distributed, results are presented with median and interquartile range [IQR] values [45]. Binary symptom variables are presented as incidence in patients who survived or died. Odds ratios and p-values for binary variables were calculated from univariate regression coefficients. For continuous

outcome variables, odds ratios are reported for a five-year increase in age and for an increase in Ct by IQR.

## Multiple imputation

In the West Africa data, 10.7% of values were missing for 16 of 18 predictor variables, and multiple imputation was used to address missing data. Details of the imputation protocol are provided in S1 Text, S1 Fig, and S1 Table.

## Variable selection

Eighteen candidate predictors including age, sex, Ct value, and 15 other epidemiological and clinical variables based on the current WHO criteria for identifying suspected Ebola cases were selected for inclusion in the model [19, 42]. These variables included fever, headache, respiratory distress (defined as fast respiratory rate; nasal flaring, grunting, intercostal recession and tracheal tug; in-drawing of lower chest wall; central cyanosis of lips and tongue; inability to breastfeed or drink; lethargy), bone or muscle pain, joint pain, conjunctivitis, asthenia, abdominal pain, hiccups, unexplained bleeding, vomiting, diarrhea, nausea, anorexia, or dysphagia [19,42]. The limit for number of candidate predictors for variable selection was set to $p < m/15$, with m being the limiting sample size equal to the minimum number of observed cases or non-cases [46].

For variable selection, we opted to use Elastic Net regularization, which combines the Lasso and Ridge regression methods, and is effective in handling multicollinearity [47]. The variable selection protocol worked as follows: Elastic Net was applied to each imputed dataset, the sign of the coefficients of the binary symptom variables in the resulting models were tallied, and those variables with the percentage of positive model coefficients above a given threshold were selected. This selection criterion facilitated the inclusion of groups of correlated predictors and predictors with small but significant effects [48]. The threshold for variable inclusion was set at 100% to exclude variables with weak and/or inconsistent effects (S2 Table).

## Model development and performance

A saturated model was constructed to serve as a baseline against which to compare the performance of other predictive models. The model included age and Ct value as continuous predictors along with four binary symptom variables selected with the Elastic Net as described above. Bootstrap resampling was used for internal validation. Discrimination was evaluated by optimism-corrected area under the receiver operating characteristic (ROC) curve (AUC) and calibration by a calibration plot comparing predicted with observed probabilities of a binary outcome, in this case survival and death [49]. The ROC curves were generated using the pROC package [50].

## External validation

We applied the West Africa derived model to the DRC data comprising 74 cases and evaluated discrimination and calibration with the optimism-corrected AUC and calibration plot. We only used cases with complete data for model validation.

## Exploratory data analysis

To further improve model performance, we followed the model recalibration with a previously outlined extension protocol [51]. We sought to add an additional biomarker as a potentially strong predictor that was available in the external validation data but not in the development

data. We focused on commonly used biochemistry laboratory values recorded within 48 hours of admission and selected covariates that were found to be significantly correlated with adverse outcome by Spearman's rank correlation coefficient. We then recalibrated the model and added an additional predictor simultaneously by fitting a new model with the linear predictors of the original model and the additional biomarkers. We did not impute missing data at this step; updated models were fit only on complete cases. Models were evaluated by comparing their AUCs and 95% confidence intervals.

## Results

### Baseline characteristics of the West Africa dataset

The West Africa dataset included 579 Ebola-positive patients less than 18 years of age with an overall CFR of 40%. Age was among the strongest predictors of mortality with each five-year decrease in age associated with an increase in the odds of death by more than half (Table 1 and S2 Fig). A Ct value below 21 was associated with higher mortality at all ages. Variables that were associated with significantly increased odds of survival included asthenia/weakness, headache, and abdominal pain (p < 0.02 each). In contrast, the presence of bleeding within the first 48 hours of admission increased the odds of death by almost 70% (p < 0.03). While the geographical distribution of cases within West Africa did not reveal a trend in CFR by location (Fig 1), there was a modest inverse correlation (r = -0.51) between CFR and number of cases at each ETU, suggesting that patients at treatment centers that had larger numbers of cases may have had less lethal outcomes for reasons that have not been determined.

**Table 1. Demographic and clinical characteristics of patients in the West Africa derivation cohort.**

| Characteristic | Survived, n = 345 | Died, n = 234 | OR (95% CI)[a] | p-value[b] |
|---|---|---|---|---|
| **Demographics** | | | | |
| Age (years), median (IQR) | 11 (7, 14) | 6 (3, 13) | 0.55 (0.46–0.65) | **<0.001** |
| Male sex, n (%) | 159 (46) | 112 (48) | 1.07 (0.77–1.5) | 0.67 |
| **Symptoms[c]** | | | | |
| Asthenia | 297 (88) | 171 (74) | 0.39 (0.25–0.6) | **<0.001** |
| Headache | 203 (60) | 93 (42) | 0.48 (0.34–0.68) | **<0.001** |
| Abdominal pain | 165 (50) | 78 (36) | 0.57 (0.4–0.81) | **0.002** |
| Bleeding | 43 (14) | 45 (22) | 1.68 (1.06–2.67) | **0.027** |
| Joint pain | 113 (38) | 48 (29) | 0.68 (0.45–1.01) | 0.060 |
| Bone or muscle pain | 121 (37) | 61 (29) | 0.71 (0.48–1.02) | 0.067 |
| Respiratory distress | 26 (7.9) | 27 (13) | 1.69 (0.95–2.99) | 0.071 |
| Vomiting | 206 (61) | 121 (54) | 0.76 (0.54–1.07) | 0.12 |
| Nausea | 172 (60) | 104 (54) | 0.78 (0.54–1.13) | 0.19 |
| Conjunctivitis | 47 (18) | 20 (15) | 0.8 (0.45–1.4) | 0.45 |
| Diarrhea | 187 (56) | 125 (59) | 1.13 (0.8–1.61) | 0.48 |
| Hiccups | 24 (7.3) | 12 (5.7) | 0.77 (0.37–1.55) | 0.48 |
| Fever | 298 (88) | 199 (87) | 0.88 (0.54–1.46) | 0.63 |
| Anorexia | 225 (74) | 128 (75) | 1.10 (0.72–1.70) | 0.67 |
| Swallowing problems | 60 (18) | 40 (19) | 1.01 (0.64–1.57) | 0.97 |
| Ct value, median (IQR) | 26.8 (23.6, 30.8) | 21.7 (18.9, 26.5) | 0.35 (0.23–0.51) | **<0.001** |

[a]OR is for each 5-year increase in age

[b]Bold values denote statistical significance

[c]n (%)

Abbreviations: IQR: interquartile range; Ct: cycle threshold; OR: odds ratio; CI: confidence intervals

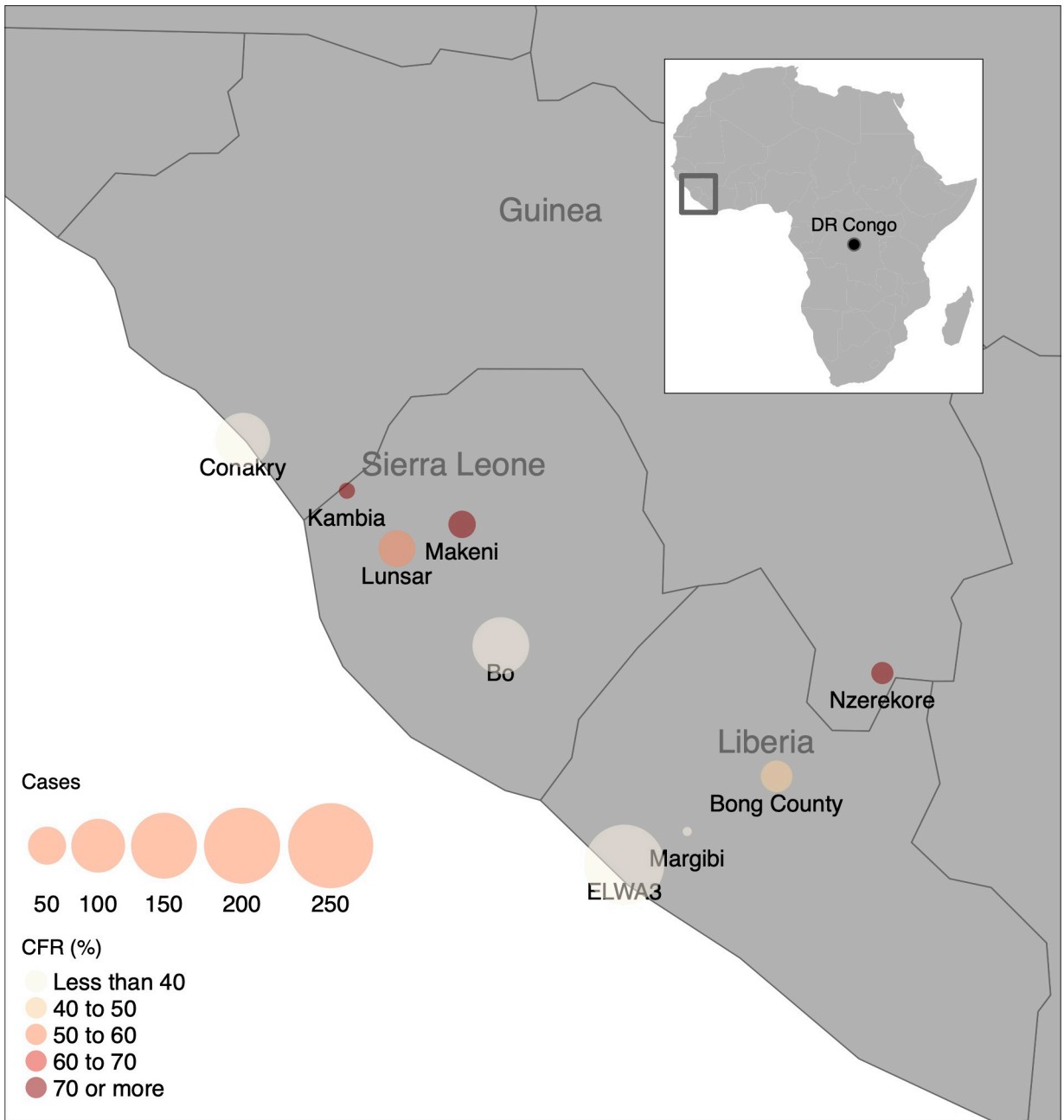

**Fig 1. Map of children with Ebola Virus Disease (EVD).** The map shows the geographical distribution of children with EVD included in triage data from the Ebola Data Platform, collected during the West African EVD outbreak from 2014–2016. Bubble size corresponds to the number of cases reported, and color corresponds to observed case fatality rate. Plotted with the R package tmap [52], using base layer maps in the public domain from the Natural Earth project (https://www.naturalearthdata.com/about/terms-of-use/).

### Derivation of clinical prognostic model

The clinical prognostic model, which we refer to as the EVD Prognosis in Children (EPiC) model, included two continuous predictors (age and Ct value) and four binary covariates (bleeding, diarrhea, respiratory distress, dysphagia). The EPiC model showed strong performance upon internal validation with AUC = 0.77 (95% CI: 0.74–0.81).

**Table 2. Comparison of baseline characteristics in West Africa derivation and DRC validation cohorts.**

|  |  | Derivation Cohort | Validation Cohort |
|---|---|---|---|
| Case-fatality rate (n, %) |  | 234 (40.4) | 22 (29.7) |
| **Continuous predictors (median, IQR)** |  |  |  |
| Age |  | 10 (5–14) | 5 (1.5–14) |
| Ct value |  | 25.1 (20.9–29.5) | 19.3 (17.6–26.1) |
| **Binary symptoms (n, %)[a]** |  |  |  |
| Bleeding |  | 88 (15.1) | 17 (22.9) |
| Diarrhea |  | 57 (9.8) | 40 (54.1) |
| Respiratory distress |  | 9.7 (1.7) | 16 (21.6) |
| Dysphagia |  | 19 (3.3) | 16 (21.6) |

[a]Covariates presented are those included in the EPiC model.

Abbreviations: IQR: interquartile range; Ct: cycle threshold

We sought to improve model performance by recalibrating the intercept and slope of the calibration plot and adding a biomarker to the model that was only available in the DRC data. An analysis of peak laboratory test results measured within the first 48 hours after admission identified three variables each significantly (p <0.01) correlated with mortality: ALT (r = 0.57), AST (r = 0.56), and CK (r = 0.51). We omitted ALT because it is highly colinear with AST (Pearson correlation = 0.83. Despite limited availability of test results in the validation data (AST: n = 29; CK: n = 33), we used these new variables to build additional models. Models that incorporated an additional predictor outperformed the original EPiC model on the validation data, in which adding CK as a predictor produced an AUC of 0.87 (95% CI: 0.74–1) while adding AST gave an AUC of 0.90 (95% CI: 0.77–1). We also considered a third model with both AST and CK added as predictors, since the association between these two biomarkers was moderate (Pearson correlation = 0.52), suggesting that they contain some amount of mutually independent information that could be combined to improve the predictions. Indeed, the model with AST and CK yields a higher AUC of 0.95 (95% CI: 0.86–1). The confusion matrix for this model exhibits an almost perfect discriminative capability with only 1 misclassification in each outcome category (S5A and S5B Table). However, the sample size for this model was reduced further to n = 23, since it requires patients to have data for both biomarkers. The ROCs and calibration plots for these three models are shown in Fig 3.

## External validation of clinical prognostic model

The DRC Mangina dataset consisted of 74 children with EVD (S2 Fig and S3 Table). A comparison of the derivation and validation cohorts is given in Table 2. The AUC on the external validation cohort was 0.76 (95% CI: 0.64–0.88) (Fig 2A). To quantitatively assess the predictive value of the EPiC model, we considered the slope and intercept of the linear fit of the calibration data and compared it against the ideal: a slope of 1 and an intercept of 0 (Fig 2B). The slope was 0.89 and the intercept -0.09, indicating that the EPiC model provides a good risk estimation overall, with only a small bias towards overestimating risk of death for all patients (except for one outlier point corresponding to a single high-risk patient) by approximately 0.1 in average. The confusion matrix and additional performance measures (including false alarm and miss rates) calculated at the optimal prediction cutoff for accuracy ($p_{cutoff}$ = 0.63) are presented in S4A and S4B Table. The $p_{cutoff}$ value of 0.63 is also consistent with the bias of around 0.1 towards risk overestimation observed in the calibration plot. Prior prognostic models [22–25] are not pediatric specific and may use different features, which makes comparisons difficult. However, we were able to apply the minimal (age+CT) model from [25], which was trained on all the patients (pediatric and adult) from the IMC ETUs in the West African EVD outbreak (a subset of the EDP dataset), on the DRC dataset. The performance is shown in S4 Fig, which shows a similar AUC of 0.77 (95% CI: 0.65–0.88), but a worse calibrated model,

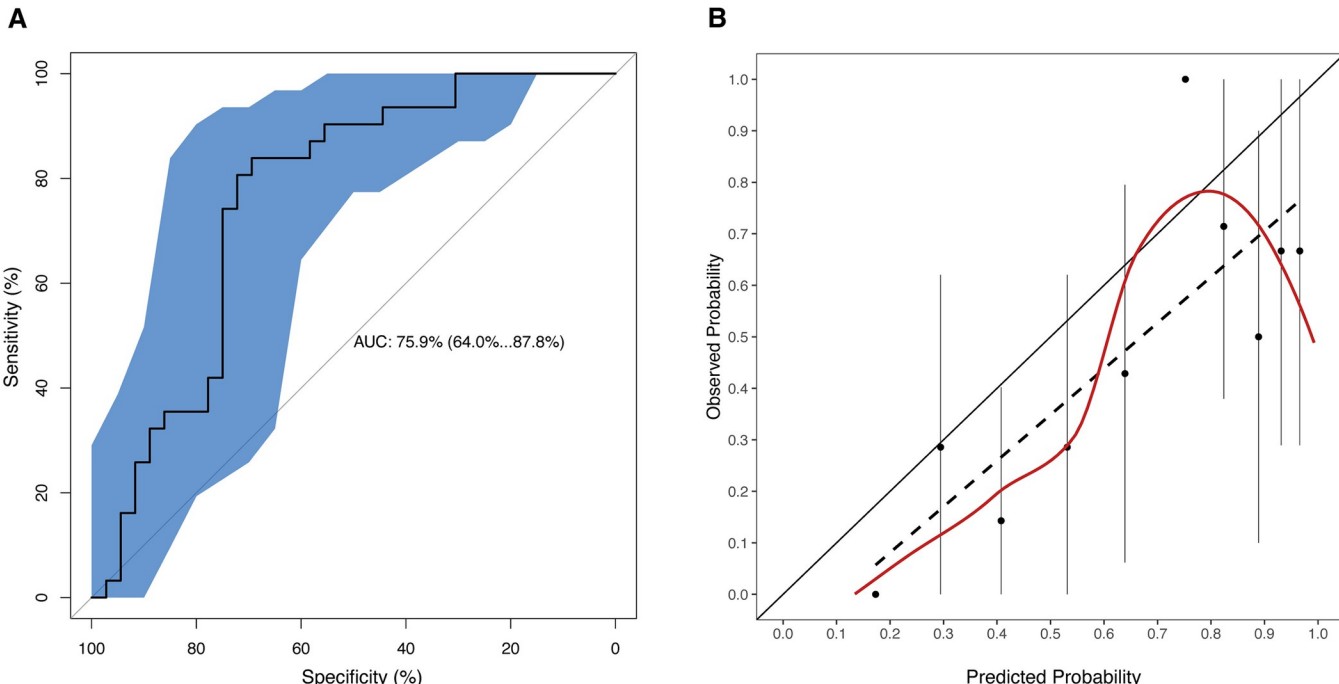

**Fig 2. Performance characteristics of the prediction model.** Discrimination (A) and calibration (B) plots of the Ebola Virus Disease Prognosis in Children (EPiC) model are shown for the Democratic Republic of the Congo validation dataset. In the discrimination plot, the receiver operating characteristic (ROC) curve is plotted (central black line) together with the 95% confidence interval band (blue shaded area). In the calibration plot, the dots represent the mean estimate of the observed probability for each 10% bin of predicted probability (with probability being risk of death), the vertical lines passing through each dot are the corresponding confidence intervals for the observed probability, the dashed line is the best linear fit passing through the mean values, and the red line is the LOESS curve fitting all the individual observed/predicted pairs in the data.

with a slope of 1.22 and an intercept of -0.27 in the linear fit to the calibration data. This is consistent with previous observations [25] that clinical features make a small contribution to the prediction relative to age and viral load, as it can be seen for our model in the ANOVA and odd ratios (OR) charts in S5 Fig. But inclusion of selected clinical features does consistently (as in our and prior studies) result in better calibrated models.

## Discussion

In this study, we derived and externally validated a prognostic model for pediatric EVD. Our model showed that younger age, lower Ct values and bleeding are poor prognostic factors while asthenia, headache and abdominal pain predict better outcomes. A few studies have described key predictors of EVD mortality among children under 18 years of age during the 2014–2015 Sierra Leone outbreak [6,18]. Shah et al reported fever, vomiting, and diarrhea as significant symptoms associated with death in children under 6 years, and Kangbai et al found that males younger than 16 years of age, who had abdominal pain, vomiting, conjunctivitis, and difficulty breathing at admission, had increased odds of dying [6,18]. A similar study in the 2014–2015 Guinea outbreak determined that older children with diarrhea, fever, and hemorrhage were at greater risk of death, while another study during the same outbreak did not report any significant risk factors for mortality among patients under 20 years [4,11]. Such findings illustrate that predicting outcomes for children with EVD presents unique challenges because the epidemiology and complications of EVD in one outbreak may vary from those in another outbreak due to differing health seeking behaviors, viral dynamics, medical interventions, and socioeconomic, cultural, and political contexts.

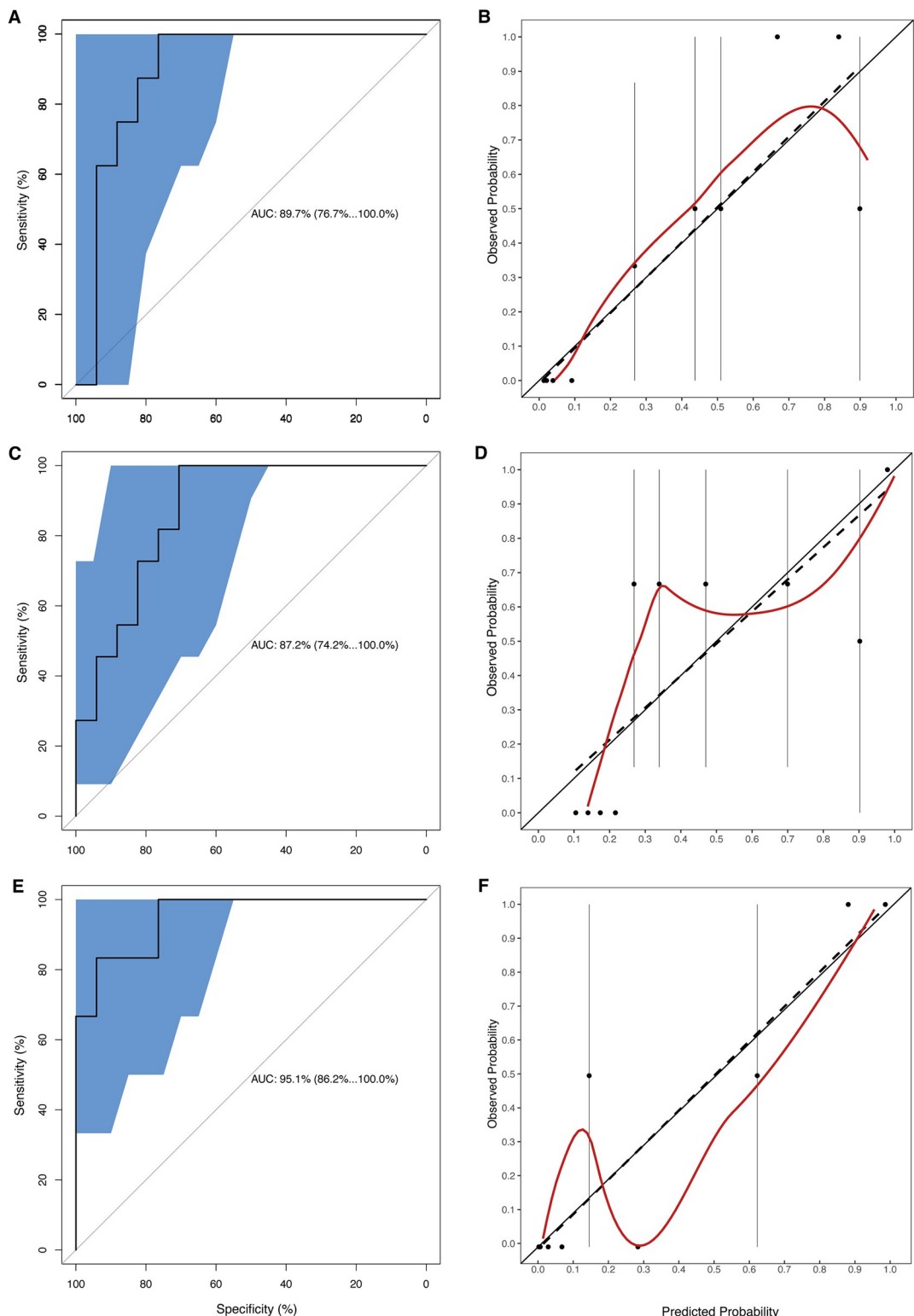

**Fig 3. Discrimination and calibration curves.** Area under the receiver operating characteristic curves (AUC) (A, C, E) and calibration curves (B, D, F) of the Ebola Virus Disease Prognosis in Children (EPiC) model are shown with aspartate aminotransferase (AST) (A, B), creatine kinase (CK) (C, D), or both (E, F) as additional predictors for the Democratic Republic of the Congo validation dataset. The interpretation of the plots is the same as in Fig 2.

Updating the EPiC model with certain biochemical tests (AST, or CK) improved its performance characteristics by a substantial margin, even though the sample size was small. AST previously has been shown to be significantly elevated in patients with EVD and associated with more severe and fatal disease [23,53,54]. Elevated AST likely reflects not only viral-induced hepatitis but also damage to other cells and end-organs such as red blood cells, pancreas, muscle, or kidneys. CK to our knowledge has not been previously described as a predictive biomarker for EVD outcomes. These biomarker data can be useful in helping to predict mortality for pediatric patients with EVD. For instance, shock may lead to an increase in AST/ALT and CK. However, the results must be interpreted with caution due to the small sample size.

The EPiC model building approach was based on Elastic Net, a form of regularized regression that has been benchmarked favorably against the more commonly-used stepwise regression [55,56]. Regularized regression is particularly good at retaining explanatory variables while reducing model complexity by removing nuisance variables. Our final EPiC model that emerged from the Elastic Net-based variable selection protocol is parsimonious in its complexity and the included predictors of EVD severity match clinical intuition. Furthermore, we were able to easily extend this protocol to update the model with additional biochemical predictors available in the DRC data. These compelling results suggest that our variable selection and model update protocol could be applied to other similar datasets.

A limitation of our study was the moderate amount of missing data (approximately 10%) for some variables, which highlights the difficulty of collecting data during a humanitarian emergency. Also, some patients may have been given experimental treatment under compassionate use, but such detailed information is not available in the West Africa derivation dataset. Furthermore, only aggregated data by day was available. As such, it was not possible to determine whether a patient died immediately upon arrival or later that day, requiring us to exclude all patients who died with one day of admission from our prediction model. The good overall calibration of our model suggests that such exclusion did not significantly affect the predictions. Additionally, our derivation dataset was collected from several different humanitarian agencies with differing data collection and laboratory procedures. Therefore, the scale of Ct values may vary between various laboratories. All Ct values presented in this manuscript were used to derive, validate, and update the models without any sort of normalization to account for the potential differences across the laboratories in the EDP dataset. Rerunning the calculations with normalized Ct values (obtained by subtracting the mean and dividing by the standard deviation at each site) revealed that all AUC values remained the same except for the AUC value on the validation dataset which was slightly lowered from 0.76 (CI 0.64–0.88) to 0.71 (CI 0.59–0.84). This indicates that the effect of Ct differences across sites is not large but also that models could be improved if raw Ct data were more consistent, or a more rigorous inter-site normalization protocol could be defined. In addition, our validation dataset is small (74 cases total) due to inclusion of only those cases with complete data, so study results have to be interpreted with caution, particularly those from model updating, which further reduced the sample size. However, these favorable preliminary results provide compelling justification for future prospective studies to investigate the prognostic utility of certain biomarkers for children as well as adults. These biomarkers, which are often part of a standard blood chemistry panel, are more accessible in low resource settings than more expensive testing such as proinflammatory cytokines [57]. Furthermore, collecting symptom information from children is difficult, especially from those who have not developed verbal skills. In fact, upon further testing, we found bone and muscle pain, asthenia, headache, and abdominal pain to be correlated with age, illustrating that children in the pre-verbal age group (defined as <2 years of age) cannot reliably report these symptoms (S3 Fig). Lastly, both settings adhered to WHO treatment guidelines and each country's respective national guidelines. As such, there may

have been slight differences in the treatment protocol between the West Africa derivation cohort and the DRC validation cohort.

In conclusion, the EPiC model is the first externally validated model for the prognosis of pediatric EVD. Pediatric patients with asthenia/weakness, headache, and abdominal pain were more likely to survive, while younger children, children with lower Ct values, bleeding, diarrhea, respiratory distress, dysphagia were more likely to die from EVD. As Ct value is a strong clinical predictor, rapid molecular tests should be widely available. The addition of routine blood test biochemical markers, such as AST and CK, strengthened the model and are usually available. This model can be easily applied by bedside clinicians to assess pediatric patients at risk for death and help to allocate resources accordingly. In fact, an online calculator has been developed so that clinicians can conveniently use the EPiC model to calculate risk scores, available at: https://kelseymbutler.shinyapps.io/epic-calculator/. Future improvements of this model would result from larger sample sizes with more consistent variable definitions and protocols across sites.

## Supporting information

**S1 Data. Excel spreadsheet containing all patients less than 18 years of age who presented to the Mangina ETU in DRC from December 2018 to January 2020 with laboratory confirmed EVD.**
(XLSX)

**S1 Text. Methods appendix on summary and imputation of missing data.**
(DOCX)

**S1 Fig. Distribution of imputed CT values for patients who presented with or without bleeding upon admission.**
(TIF)

**S2 Fig.** Flowchart of patients excluded in West Africa Derivation cohort (A) and DRC validation cohort (B).
(TIF)

**S3 Fig. Correlation between bone and muscle pain, asthenia, headache, and abdominal with age group.** Age group was defined as pre-verbal (<2 years), early verbal (2-<5 years), verbal (5-<10 years), verbal (10–17 years).
(TIF)

**S4 Fig.** Receiver operating characteristic (ROC) (A) and calibration (B) plots for the minimal model (age+CT) described in [22], trained on all patients (not pediatric-specific) in the EVD West African dataset from IMC. The intercept and slope of the linear fit to the predicted probabilities are -0.27 and 1.22, respectively.
(TIF)

**S5 Fig. Plots showing the importance of the features in the EPiC model.** Analysis of variance chart generated with the anova() function in the rms package, showing a ranking of the features according to their predictive contribution to the model, as measured by the Wald $\chi^2$-d.f. (degrees of freedom) statistic (A). Chart generated with the summary function in rms, showing the odds ratios for all the features in the model, using interquartile-range odds ratios for continuous features (age and CT), and simple odds ratios for binary (yes/no) features (B).
(TIF)

**S1 Table. Summary of missing values in West Africa derivation dataset.**
(XLSX)

**S2 Table. Results of variable selection protocol for categorical variables.** Variables were included only if they were selected in 100% of the models, which includes bleeding, diarrhea, respiratory distress, and swallowing problems.
(XLSX)

**S3 Table. Demographic and clinical characteristics of patients in DRC validation cohort.**
(XLSX)

**S4 Table.** Confusion matrix (a) and detailed performance measures (b) of the EPiC model as applied on the validation DRC dataset at the optimal prediction cutoff for accuracy (0.63). Both tables are the result of the confusionMatrix() function in the R package Caret (https://cran.r-project.org/web/packages/caret/index.html), with the addition of the false alarm and miss rates to S4B Table, and removal of a few measures that are less commonly used (e.g., Mcnemar's Test P-Value and Kappa coefficient).
(XLSX)

**S5 Table.** Confusion matrix (a) and detailed performance measures (b) of the EPiC model augmented with AST and CK as applied on the validation DRC dataset at the optimal prediction cutoff for accuracy (0.5). Interpretation of these tables is the same as in S4 Table.
(XLSX)

## Acknowledgments

The authors would like to thank the International Medical Corps field teams who serve tirelessly to provide excellent care to patients with Ebola Virus Disease, the Advance Clinical and Translational Research (Advance-CTR) at Brown University, and the staff of the Infectious Diseases Data Observatory Ebola Data Platform without whom this study would not be possible. The content of this manuscript is solely the responsibility of the authors and does not necessarily represent the views of any governmental bodies or academic organizations.

## Author Contributions

**Conceptualization:** Alicia E. Genisca, Kelsey Butler, Adam C. Levine, Ian C. Michelow, Andrés Colubri.

**Data curation:** Monique Gainey, Tzu-Chun Chu, Lawrence Huang, Eta N. Mbong, Razia Laghari, Fiston Nganga, Rigobert F. Muhayangabo, Himanshu Vaishnav, Shiromi M. Perera.

**Formal analysis:** Alicia E. Genisca, Kelsey Butler, Ian C. Michelow, Andrés Colubri.

**Writing – original draft:** Alicia E. Genisca, Kelsey Butler, Monique Gainey, Ian C. Michelow, Andrés Colubri.

**Writing – review & editing:** Alicia E. Genisca, Kelsey Butler, Monique Gainey, Tzu-Chun Chu, Lawrence Huang, Eta N. Mbong, Stephen B. Kennedy, Razia Laghari, Fiston Nganga, Rigobert F. Muhayangabo, Himanshu Vaishnav, Shiromi M. Perera, Moyinoluwa Adeniji, Adam C. Levine, Ian C. Michelow, Andrés Colubri.

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
