## [Decision Letter · Decision Letter 0]

19 Apr 2022

Dear Dr Colubri,

Thank you very much for submitting your manuscript "Using machine learning to predict survival in children with Ebola Virus Disease" for consideration at PLOS Neglected Tropical Diseases. As with all papers reviewed by the journal, your manuscript was reviewed by members of the editorial board and by several independent reviewers. In light of the reviews (below this email), we would like to invite the resubmission of a significantly-revised version that takes into account the reviewers' comments. 

We cannot make any decision about publication until we have seen the revised manuscript and your response to the reviewers' comments. Your revised manuscript is also likely to be sent to reviewers for further evaluation.

Sincerely,

Anita K. McElroy, MD, PhD

Associate Editor

Camille Lebarbenchon

Deputy Editor

Reviewer's Responses to Questions

**Key Review Criteria Required for Acceptance?**

**Methods**

-Are the objectives of the study clearly articulated with a clear testable hypothesis stated?

-Is the study design appropriate to address the stated objectives?

-Is the population clearly described and appropriate for the hypothesis being tested?

-Is the sample size sufficient to ensure adequate power to address the hypothesis being tested?

-Were correct statistical analysis used to support conclusions?

-Are there concerns about ethical or regulatory requirements being met?

Reviewer #1: -There are several limitations to the methods. Many of the data elements were missing- in fact 10% of data were missing. 

Additionally there was a variety in the manner in which the patients were treated and data were collected. However, given the circumstances it is difficult to envision how the data could have been collected differently during the epidemic. The authors should mention which of the 18 variables were missing and in what percentages in the Supplemental material

-It would help if the authors provided more details about the IDDO EDP. Were data collected retrospectively or prospectively. 

-suggest providing more details about how information about symptoms were collected

-suggest specifying why a 5 year age gap was used? did the authors discuss dividing age into a categorical variable since a 4 year old is very different from an infant? Perhaps they could do a sensitivity analysis using age groups that are categorical (infant, toddler, etc.). Would also suggest including more details about the number of patients in each categorical age group. The IQR is presented but it would be helpful to have more details

-please cite the reference for lines 156 regarding the Shapiro Wilk test

Reviewer #2: Using machine learning to predict survival in children with Ebola Virus Disease

Alicia E. Genisca, et al.

While Ebola virus disease (EVD) is well known to cause a highly lethal disease in all age groups, it is especially lethal young, pediatric patients. In this report, Genisca and colleagues use a machine learning approach to develop a prognostic model called the EVD prognosis in Children (EPiC). The initial EPiC model was created using a training dataset created using information based upon pediatric patient in the 2014-16 West Africa EVD outbreak. This resulted in a model that used age, Ct value, bleeding, diarrhea, breathlessness and dysphagia. This initial model performed with an AUC of 0.77 in the training cohort and when evaluated using a smaller validation cohort from the Democratic Republic of Congo (DRC) showed a similar performance. Further evaluation of the model in the validation cohort found that it tended to overestimate the risk of death. Since this performance was not optimal, two additional laboratory parameters, AST and CK, were investigated to determine whether adding their values to the model would improve performance. They found the predictive AUC was 0.87 to 0.90. Overall, the claims of this paper are supported by the data presented. However, parameters that were identified were already well known to be linked to poor outcomes and the predictive model was not translatable into a clinically useful tool that would allow physicians to easily use it to cohort patients based on risks. 

Comments: 

1) Line 121-123: Patient selection excluded subjects if they were missing outcome data or if they died within 24 hours of admission to the Ebola treatment unit, “as a prediction tool would not be useful for a moribound patient”. 

• A flow chart/decision tree for each cohort that shows how many patients were excluded for each reason should be provided as a supplemental figure. This will help assess for any potential biases in their training and validation cohorts.

• This reviewer respectfully disagrees with the assumption that death within 24 hours is always going associated with the patient being moribound at the time of admission or that there might not be a reversable condition that could be addressed if appropriate risk factors were identified early. This is because interventions could be targeted by clinicians to address the risk factors. 

2) While, the timing of the ALT and CK sample values used were indicated, it was not clear what time point was used to make the determination for the parameters evaluated in the models. That is, was it at admission, if the symptom developed at any point or something else. Similarly, for Ct, was the value at admission, the peak value or something else? 

3) Were there differences in clinical site PCR testing in terms of the viral genes targeted and specific test that determined the Ct value? If so, how were differences accounted for in the model?

4) In the West African outbreak, some patient were able to receive experimental therapies under compassionate use, expanded access or as part of clinical trials. It would be helpful to see the number of patients in each group that received an experimental therapy, especially monoclonal antibodies.

5) The manuscript notes that the original model overestimated the risk of death in the DRC validation cohort, this could be due to a larger number of patients in the validation cohort receiving effective therapeutics as part of expanded access (EUA) protocols or as part of the PALM randomized clinical trial. Particularly, two of the therapies were shown to be highly effective at reducing mortality. A supplementary table should be provided that shows the complete set of clinical parameters shown in Table 1 for the DRC cohort and that shows the number of patients in the cohort that received an experimental therapy under EUA or the PALM randomized clinical trial. 

6) The results section discussing the Derivation of the Clinical Prognostic Model (lines 231-235) states what parameter were used in the model but does not explain to the reader how those were chosen amongst all the significantly different variables shown in Table 1. The manuscript would be improved by adding a few more details on how the machine learning protocol reduced parameter sets to two continuous and 4 binary covariates.

7) Line 252-253 indicated that ALT was not used in the models and that ALT or CK were. This was because ALT was highly correlated with AST. Therefore, all comments and statements in the manuscript that ALT was used should be removed. For example on lines 280.

8) Line 312-313. It is not clear from this report how clinician would use this model to predict risk because there is not a toolkit online or clinical scoring system or something similar that one would utilize to calculate a risk. Also, the formal definitions of how the clinical parameters used were defined is not available. For example, what is the formal definition of breathlessness? Is it a subjectively determined or is it an abnormal respiratory rate or is it having signs of severe respiratory distress or something else. Thus, statement about ease of application of the mode should be removed.

Minor comments

9) Line 239. Validation AUC is 0.79 on this line and in the abstract is it 0.76

10) Figure 2B. The figure would be improved by providing a description in the legend or by labels in the panel for what the dashed and red lines are.

Reviewer #3: - It is not clear why there is a need for a new model. As also the authors state in the manuscript, there are existing predictive models. These models were not evaluated using training and validation sets separately. The same datasets could be used to train, test and validate the existing predictive models. At least to justify the need for a new predictive model the proposed model could be compared to these existing models over the same dataset. 

- Bootstrapping is used for model training (or derivation t as authors denote in the manuscript). This approach could overestimate the training performance. Why not use K-fold cross validation? 

- It will be good to prepare a table that includes all the variables that are used in Elastic net for variable selection. Also this table could include information about if these variables are continuous valued or binary

**Results**

-Does the analysis presented match the analysis plan?

-Are the results clearly and completely presented?

-Are the figures (Tables, Images) of sufficient quality for clarity?

Reviewer #1: The data presented do match the analysis plan and the figures are legible and clear. A few minor suggestions:

-In Table 1, I would add a footnote mentioning that the OR is for each 5 year increase in age

-Suggest a new title for Table 1 reflecting that this title is focused on mortality and predictors of mortality

-The mortality, location, and age of the validation cohort appear to be very different from the original cohort. Do the authors think this might affect their results?

Reviewer #2: see above

Reviewer #3: - It is not clear how many patients were in the training and how many were in the validation? Tables 1 and 2 are confusing, as the number participants used for derivation in Table 2 is 234 while total number of patients in external validation set is 74. 

- What does derivation cohort presented in Table 2 mean? Is the derivation cohort from DRC Mangina dataset used to refine the model trained using the data described in Table 1? If yes, the results do not present an external validation. Evaluation data should not be used in the training. 

- Overall description of results need to be more clear, more clear description of Tables are needed. 

- Why not train and test the existing predictive methods using the same datasets to compare with the proposed method? This will also justify the need for a new predictive model. 

- How are the Sensitivity vs Specificity curves obtained? This needs be explained clearly. Also what are the probability of false alarm and probability of miss?

**Conclusions**

-Are the conclusions supported by the data presented?

-Are the limitations of analysis clearly described?

-Do the authors discuss how these data can be helpful to advance our understanding of the topic under study?

-Is public health relevance addressed?

Reviewer #1: -the authors did not include treatment type or type of facility in their analysis. Do the authors think this could be a potential limitation of their study?

-why do the authors think that asthenia, headache, and abdominal pain were correlated with better outcomes? could this have to do with the ability of the child to relate their symptoms to a caregiver? in other words a child with a better mental status or who is older may be better able to express those symptoms?

-authors should mention that collecting symptom information in children is difficult and may be a limitation (especially for very young children)

-line 283- the authors should mention specifically that shock may lead to an increase in AST/ALT and CK

-another limitation that the authors should mention is that Ct values vary from assay to assay

-the validation model only looked at 74 cases in a different setting and time frame which is an additional limitation

Reviewer #2: see above

Reviewer #3: (No Response)

**Editorial and Data Presentation Modifications?**

Reviewer #1: (No Response)

Reviewer #2: (No Response)

Reviewer #3: (No Response)

**Summary and General Comments**

Reviewer #1: This is a well written manuscript that provides a model for assessing risk of death in pediatric patients with EVD. This is an important study because it can prove helpful in future epidemics. It would be further strengthened if it included data on therapeutics and care.

Reviewer #2: (No Response)

Reviewer #3: (No Response)

PLOS authors have the option to publish the peer review history of their article (what does this mean?). If published, this will include your full peer review and any attached files.

Reviewer #1: No

Reviewer #2: No

Reviewer #3: No
---

## [Decision Letter · Decision Letter 1]

28 Jul 2022

Dear Dr Colubri,

Thank you very much for submitting your manuscript "Constructing, validating, and updating machine learning models to predict survival in children with Ebola Virus Disease" for consideration at PLOS Neglected Tropical Diseases. As with all papers reviewed by the journal, your manuscript was reviewed by members of the editorial board and by several independent reviewers. The reviewers appreciated the attention to an important topic. Based on the reviews, we are likely to accept this manuscript for publication, providing that you modify the manuscript according to the review recommendations. 

Sincerely,

Anita K. McElroy, MD, PhD

Academic Editor

Camille Lebarbenchon

Section Editor

Reviewer's Responses to Questions

**Key Review Criteria Required for Acceptance?**

**Methods**

-Are the objectives of the study clearly articulated with a clear testable hypothesis stated?

-Is the study design appropriate to address the stated objectives?

-Is the population clearly described and appropriate for the hypothesis being tested?

-Is the sample size sufficient to ensure adequate power to address the hypothesis being tested?

-Were correct statistical analysis used to support conclusions?

-Are there concerns about ethical or regulatory requirements being met?

Reviewer #2: see editorial and date presentation section

Reviewer #3: The authors addressed most of our concerns. However, we still think that a comparison with the existing methods will motivate the need for a new predictive model better. The authors claim that the previously developed predictive models are limited in geographic and temporal scope. Our understanding is that the previous models while being trained and tested did not have access to the datasets that the authors used in this manuscript. When trained and tested in the same dataset, these existing models may outperform the methodology that the authors are proposing in this paper. We think that the authors need to demonstrate that the predictive model they are proposing is outperforming the existing methods when the same datasets are used. This will provide a better motivation for the proposed methodology. Then if the proposed method outperforms the existing methods when used on the same dataset, a discussion on methodological differences and why these differences matter should be discussed.

**Results**

-Does the analysis presented match the analysis plan?

-Are the results clearly and completely presented?

-Are the figures (Tables, Images) of sufficient quality for clarity?

Reviewer #2: see editorial and date presentation section

Reviewer #3: please see above.

**Conclusions**

-Are the conclusions supported by the data presented?

-Are the limitations of analysis clearly described?

-Do the authors discuss how these data can be helpful to advance our understanding of the topic under study?

-Is public health relevance addressed?

Reviewer #2: see editorial and date presentation section

Reviewer #3: please see above.

**Editorial and Data Presentation Modifications?**

Reviewer #2: If not commented on below, the responses provided by the authors has satisfied this reviewer.

Initial Reviewer comment

Line 121-123: Patient selection excluded subjects if they were missing outcome data or

if they died within 24 hours of admission to the Ebola treatment unit, “as a prediction tool

would not be useful for a moribound patient”. A flow chart/decision tree for each cohort

that shows how many patients were excluded for each reason should be provided as a

supplemental figure. This will help assess for any potential biases in their training and

validation cohorts. This reviewer respectfully disagrees with the assumption that death

within 24 hours is always associated with the patient being moribound at the time of

admission or that there might not be a reversible condition that could be addressed if

appropriate risk factors were identified early. This is because interventions could be

targeted by clinicians to address the risk factors.

○ Thank you for your thoughtful comment. We respectfully disagree with the

reviewer. Based on the experience in the field of some of the co-authors of this

paper, we believe that inclusion of children who died within the first 24 hours

could bias the model because the definitive diagnosis may not be confirmed and

Ct values as well as most other laboratory results would not be available within

that short timeframe. Furthermore, it may not be practical or meaningful for

clinicians to apply such a model when they are focused on resuscitating an

unstable moribund patient under challenging circumstances. Also, we

constructed the model taking into account that patients may respond well to

resuscitative measures (e.g., rehydration, glucose administration, electrolyte

supplementation) within the first 24 hours, in which case, the model would not be

accurate. We concluded that inclusion of these cases may have detracted from

the model because of the large number of missing data. Therefore, taken

together, we believe that by excluding unstable moribound children, we present a

more robust and clinically meaningful model that is likely to be more accurate

and generalizable to other settings. Additionally, per the reviewer’s suggestion a

flow chart has been provided as a supplemental figure detailing how many

patients were excluded from each dataset.

Reviewer response:

o The way the section on participant Selection is written that is being discussed, it is appears to assume that someone who dies in thee first 24 hours is moribund at the time of admission. Could they not become moribund after admission but within the first 24 hours? In that case, having a tool available to predict which ones might become moribund within the first 24 hours might be useful.

That said, I would be satisfied if the authors would rewrite this sentence to be more consistent with the authors response above. 

o Thank you for providing the data on excluded patients in each cohort. The DRC validation cohort has ~24% of patients excluded, whereas as the West Africa had ~4%. This could influence the outcomes being analyzed. For example could the overestimation of death in the DRC cohort be because of the higher proportion of patient who died in the first 24 hours being excluded? Some comment should be provided in the manuscript.

Initial Reviewer comment

The results section discussing the Derivation of the Clinical Prognostic Model (lines 231-

235) states what parameter were used in the model but does not explain to the reader

how those were chosen amongst all the significantly different variables shown in Table

1. The manuscript would be improved by adding a few more details on how the machine

learning protocol reduced parameter sets to two continuous and 4 binary covariates.

○ Thank you for the feedback. The section the Reviewer is referring to is in the

Results section of the manuscript. Details on how the final variables were

selected for the model is detailed in the Methods section under the “Variable

selection” subsection. In short, the binary variable selection protocol is as follows:

Elastic Net was applied to each imputed dataset, the sign of the coefficients of

the binary symptom variables in the resulting models were tallied, and those

variables with the percentage of positive model coefficients above a given

threshold were selected (Supplementary Table 2). This selection criterion

facilitated the inclusion of groups of correlated predictors and predictors with

small but significant effects. The threshold for variable inclusion was set at 100%

to exclude variables with weak and/or inconsistent effects. Based on this

protocol, any bleeding, dysphagia, breathlessness, and diarrhea were included in

the EPiC model. Age and Ct were selected for inclusion in the model as they are

highly correlated with poor prognosis in children with (references cited below)..[...]

Reviewer response:

o Thanks for you response. I appreciate the details are in the methods. The comment was to help the reader of the manuscript better follow the logic without needing to refer back to the methods section. The recommendation stands but it is not a requirement.

Initial Reviewer comment:

● Line 312-313. It is not clear from this report how clinician would use this model to predict

risk because there is not a toolkit online or clinical scoring system or something similar

that one would utilize to calculate a risk. Also, the formal definitions of how the clinical

parameters used were defined is not available. For example, what is the formal definition

of breathlessness? Is it a subjectively determined or is it an abnormal respiratory rate or

is it having signs of severe respiratory distress or something else. Thus, statement about

ease of application of the mode should be removed.

○ Thank you for the feedback. Based on your suggestion, we developed an online

calculator so that physicians can easily use it to calculate risk scores. This

calculator can be found at https://kelseymbutler.shinyapps.io/epic-calculator/ and

a link has been included in the text: As stated in the methods section of the

manuscript, clinicians at each Ebola treatment center followed the World Health

Organization’s guidelines for clinical assessment and definitions of abnormal

signs and symptoms e.g. The WHO manual refers to age-related respiratory

rates for respiratory distress. We cited the WHO guidelines in our Reference list

(#19 and #42).

Reviewer Response:

o Thank you for providing the online calculator. 

o The link for reference 19 needs to be updated.

o While the presence of bleeding, diarrhea and dysphagia can reasonably be pulled from a chart breathlessness is more subjective. In addition, the WHO documents discussed do not contains a definition for breathlessness. Respiratory distress is defined. However, even if breathlessness was defined in the WHO documents, a reader should not have to go to the WHO guidelines to determine what was meant. Please provide the formal definitions used in your models in the manuscript.

Reviewer #3: (No Response)

**Summary and General Comments**

Reviewer #2: see editorial and date presentation section

Reviewer #3: (No Response)

PLOS authors have the option to publish the peer review history of their article (what does this mean?). If published, this will include your full peer review and any attached files.

Reviewer #2: No

Reviewer #3: No

Figure Files:

Data Requirements:

Reproducibility:

References

---

## [Editor Report · Decision Letter 2]

5 Sep 2022

Dear Dr Colubri,

We are pleased to inform you that your manuscript 'Constructing, validating, and updating machine learning models to predict survival in children with Ebola Virus Disease' has been provisionally accepted for publication in PLOS Neglected Tropical Diseases.

Best regards,

Anita K. McElroy, MD, PhD

Academic Editor

Camille Lebarbenchon

Section Editor

---

## [Editor Report · Acceptance letter]

28 Sep 2022

Dear Dr Colubri,

We are delighted to inform you that your manuscript, "Constructing, validating, and updating machine learning models to predict survival in children with Ebola Virus Disease," has been formally accepted for publication in PLOS Neglected Tropical Diseases.

Best regards,

Shaden Kamhawi

co-Editor-in-Chief

Paul Brindley

co-Editor-in-Chief
